# The conformational changes coupling ATP hydrolysis and translocation in a bacterial DnaB helicase

Thomas Wiegand[1], Riccardo Cadalbert[1], Denis Lacabanne [1,2], Joanna Timmins [3], Laurent Terradot [2], Anja Böckmann [2] & Beat H. Meier [1]

DnaB helicases are motor proteins that couple ATP-hydrolysis to the loading of the protein onto DNA at the replication fork and to translocation along DNA to separate double-stranded DNA into single strands during replication. Using a network of conformational states, arrested by nucleotide mimics, we herein characterize the reaction coordinates for ATP hydrolysis, DNA loading and DNA translocation using solid-state NMR spectroscopy. AMP-PCP is used as pre-hydrolytic, ADP:AlF$_4^-$ as transition state, and ADP as post-hydrolytic ATP mimic. $^{31}$P and $^{13}$C NMR spectra reveal conformational and dynamic responses to ATP hydrolysis and the resulting DNA loading and translocation with single amino-acid resolution. This allows us to identify residues guiding the DNA translocation process and to explain the high binding affinities for DNA observed for ADP:AlF$_4^-$, which turns out to be optimally preconfigured to bind DNA.

[1] Physical Chemistry, ETH Zurich, 8093 Zurich, Switzerland. [2] Molecular Microbiology and Structural Biochemistry, Labex Ecofect, UMR 5086 CNRS/ Université de Lyon, 69367 Lyon, France. [3] Univ. Grenoble Alpes, CNRS, CEA, CNRS, IBS, F-38000 Grenoble, France. Correspondence and requests for materials should be addressed to A.Böc. (email: a.bockmann@ibcp.fr) or to B.H.M. (email: beme@ethz.ch)

Helicases are involved in many aspects of nucleic-acid metabolism. Here we are interested in their function as part of the replisome, catalyzing the unwinding of double-stranded DNA and separation into two single-stranded intermediates (ssDNA) each of which is then complemented by a polymerase yielding identical copies of the genetic material[1–3]. The helicase loads (with or without loader proteins) at the replication fork and translocates along the DNA opening further nucleotide pairs. It can be looked at as a molecular machine fuelled by ATP which is hydrolyzed to ADP and inorganic phosphate (Pi) in the nucleotide binding domain (NBD). The NBD is the heart of the motor part driving the translocation through the contacts of the DNA-binding domains with ssDNA. This cycle is coupled to the process of DNA loading and the directional translocation of the DNA[4].

The bacterial helicase from *Helicobacter pylori* (*Hp*) belongs to superfamily 4 (SF4) helicases[5–7] and consists of two domains: The C-terminal domain (CTD) hosts the molecular motor, as well as binds and transports the ssDNA; and the N-terminal domain (NTD) binds to the DnaG primase[8,9]. Crystal structures of bacterial replicative helicases (belonging to SF4) in complex with ssDNA were obtained from *Bacillus stearothermophilus* (*Bst*DnaB)[10] and *Geobacillus kaustophilus HTA426* (*Gk*DnaC)[11]. In the structure of *Bst*DnaB bound to five molecules of GDP-AlF$_4^-$, a spiral staircase conformation was observed suggesting a hand-over-hand DNA translocation mechanism in which the subunits move along the DNA during translocation driven by a sequential ATP hydrolysis mechanism[10], in contrast to a concerted model in which ATP hydrolysis occurs simultaneously in all subunits[6]. A similar mechanism but based on an almost flat six-fold helicase structure was proposed for the E1 and Rho helicases (belonging to SF3 and SF5, respectively): the DNA-binding loops of the helicase move during translocation and transport ssDNA through the inner channel of the multimer.[4,12] For *Gk*DnaC, a flat, pseudo 6-fold symmetric structure was also observed in presence of DNA, but with three independent DNA

stretches bound within the central channel of the hexameric protein[11].

The helicase-DNA interactions must be coupled to the state of the NBD that is running through the ATP hydrolysis cycle. At least in vitro, the loading of *Hp*DnaB onto ssDNA also seems catalyzed by the same motor domain[7]. The motor domains of helicases are composed of three characteristic structural motifs that are responsible for binding the nucleotide predominantly via hydrogen bonds (see Supplementary Figure 1): the Walker A (phosphate-binding loop, residues 203A-210T for *Hp*DnaB) and Walker B motifs (coordinates Mg$^{2+}$ cofactor, residues 309I-316Q), as well as the arginine finger (R-finger, binds to the γ-phosphate, residues 445N-451G) connecting two adjacent DnaB monomers.[13]

A schematic view of the reactions being studied in the following is provided as reaction coordinates on a two-dimensional reduced energy surface[14] in Fig. 1a. The ATP hydrolysis reaction powering the motor domain of DnaB is indicated in blue; the binding to ssDNA starting from different intermediate states on the blue pathway is indicated by green reaction coordinates, which connect the blue hydrolysis trajectory with the red DNA-translocation reaction coordinate.

Interesting points in the motional processes along the reaction coordinate of ATP hydrolysis are the (i) apo state, (ii) ATP binding, (iii) breakage of the Pβ-O-Pγ bond and (iv) release of ADP and Pi. The most practical method to experimentally study the conformational states involved in these processes is to arrest the conformation to produce snapshots thereof[15]. This can be achieved by replacing the ATP nucleotide with various ATP-analogues[16] (in the following called nucleotides) mimicking the different states i-iv along the ATP-hydrolysis reaction: the apo state (no nucleotide added); a pre-hydrolytic state (employing adenylyl-methylene-diphosphonate (AMP-PCP)[17] where the Pβ-CH$_2$-Pγ bond mimics an intact Pβ-O-Pγ bond); the transition state (TS) where the Pγ subunit is replaced by AlF$_4^-$ which mimics an already weakened Pβ-O-Pγ unit[18–20]; and a post-

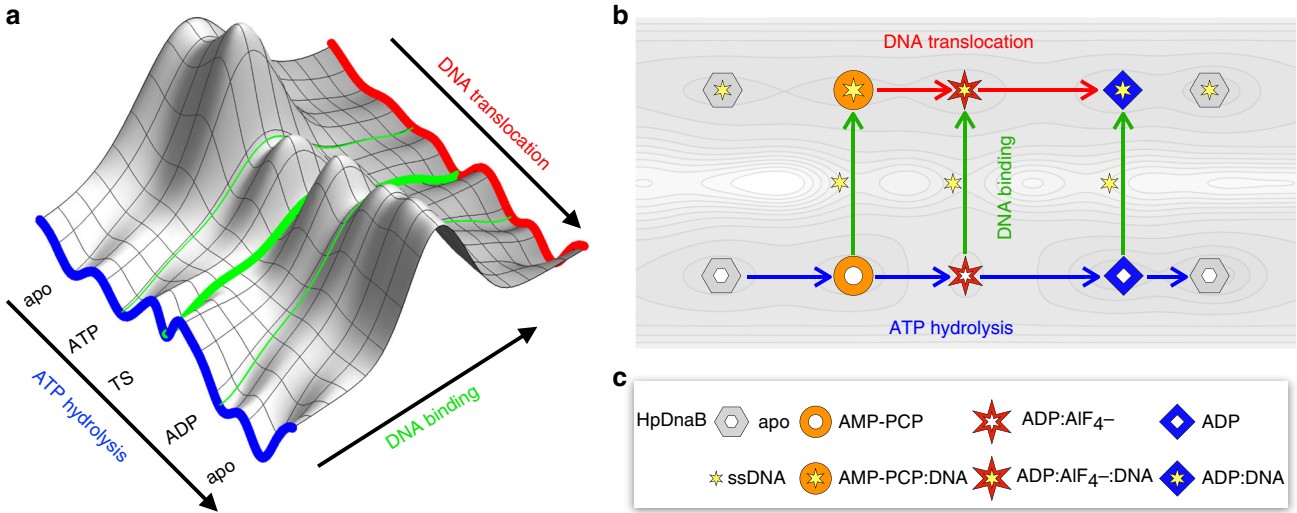

**Fig. 1** Schematic overview of DnaB reaction coordinates studied herein. Scheme of the different reactions involved in DNA binding and translocation by DnaB on a reduced schematic energy surface (**a**), and equivalence scheme where the different dynamically visited states in **a** are mimicked by stable protein states arrested by the appropriate ATP analogue (**b**). The blue reaction coordinate in **a** and the blue arrows in **b** describe the ATP hydrolysis reaction taking place in the NBD. ATP stands for DnaB:ATP before hydrolysis takes place, TS for the transition state and ADP for the post-hydrolytic DnaB: ADP + Pi state before the nucleotide is released to return to the apo state from where the reaction continues cyclically. For the mimics defining the equivalence reaction of **b** see subfigure (**c**). AMP-PCP:DNA stands for DnaB:AMP-PCP bound to ssDNA etc. The green reaction coordinates describe DNA binding starting from either the pre-hydrolytic, transition, and post-hydrolytic state, and result in a DNA-bound state, as indicated by a yellow star in the centre of the symbol. The most efficient pathway (vide infra) is indicated, in **a**, by a thick green line, the two alternatives by thin green lines. The process of DNA translocation is indicated by a red reaction coordinate and arrows, in **a, b**, respectively

hydrolytic state in which the Pβ-O-Pγ bond is broken (ADP has been generated), before the apo state is reached again (see also Supplementary Figure 1). The pre-hydrolytic state proved the most difficult to mimic. Two alternative compounds to AMP-PCP, i.e. AMP-PNP and ATPγS were tested, but both (the first only in presence of DNA[21]) were hydrolyzed during sample preparation (rotor filling).

Although the nucleotides used are chemically different from ATP, the DnaB:nucleotide complexes are assumed to represent the local minima on the ATP hydrolysis reaction coordinate labelled as apo, ATP, TS, ADP in Fig. 1a. Note the notation where we leave away the DnaB specification and use ATP instead of DnaB:ATP throughout the article. As an approximation to the fugitive states on the reaction coordinate of the hydrolysis reaction, we can now experimentally study the equivalence scheme of Fig. 1b where every symbol represents a stable compound with conformation and chemical properties similar to the equivalent elusive local minima points in Fig. 1a.

We here show, using nucleotides mimicking different stages of ATP-hydrolysis, that binding and hydrolysis of ATP lead to significant structural and, maybe even more importantly, dynamic changes for a significant, contiguous portion of DnaB. These changes are highly correlated with the location of the residues in the 3D structure, and define the transduction pathways linking ATP hydrolysis (the source of free energy) to ssDNA binding and translocation. Our study shows that most efficient DNA binding proceeds via the transition state which prepares the conformation of the protein for DNA binding. The movement of the protein along DNA occurs via few non-covalent contacts with residues in the DNA-binding loop of the helicase, and especially 357R and 373K could be involved in pulling the DNA through the inner pore, as reflected by the conformational variation during the DNA translocation process.

## Results

**ATP hydrolysis reaction cycle.** In the first part of this study, we characterize the conformational events along the equivalence scheme for the ATP hydrolysis reaction (blue arrows in Fig. 1b, for the nomenclature of the studied complexes see Fig. 1c). The four DnaB states investigated can be regarded as representing four consecutive states in the ATP hydrolysis cycle, as shown in the sketch in the upper left corner of Fig. 2.

Phosphorus-31 NMR allows to characterize nucleotide binding; [31]P spectra of the bound nucleotides AMP-PCP and ADP:AlF$_4^-$, as well as of ADP are shown in the first column in Fig. 2. These spectra were obtained with the cross-polarization (CP) technique that only shows species showing the long rotational correlation times typical for the protein immobilized by sedimentation, and not molecules in the supernatant fraction which show fast tumbling. The nucleotides are thus indeed bound to the protein. For all three states, the expected number of [31]P resonances (3 for AMP-PCP, 2 otherwise) is observed, indicating the structural equivalence of all 6 monomers of the hexamer. The [31]P resonances are the narrowest for ADP:AlF$_4^-$, indicating highest structural order and homogeneity, while for AMP-PCP, a slight broadening of the Pα and Pβ resonances is observed indicating small variations of the electronic environment of the [31]P spins or in the conformation of each nucleotide (vide infra). Addition of ATP to the apo form yields the ADP spectrum, implying that ATP is hydrolyzed to ADP and Pi is released on the timescale of sample preparation and only the ADP-bound state can thus be investigated (see Supplementary Figure 2).

Carbon-13 detected NMR spectra probe conformational and dynamic features of a protein state. The second column of Fig. 2 shows a respective region from $^{13}$C-$^{13}$C 20 ms dipolar assisted

rotational resonance (DARR)[22,23] NMR spectra for the different nucleotide-bound states (for larger spectral windows see Supplementary Figure 3). DARR is a robust and reliable experiment that allows to obtain a correlation spectrum showing crosspeaks between close-by and, at 20 ms mixing, typically neighbouring $^{13}$C nuclei (peaks that for example show in one dimension the Cα and in the second dimension the Cβ chemical shift of the same amino acid). It provides valuable spectral fingerprints, typically in the Cα/Cβ chemical-shift region, where one signal is observed per amino acid that can be used to follow conformational changes for every amino acid. Alanine signals, as they are the only amino-acid type which correlates a CH with a CH$_3$ group in the Cα/Cβ region, are generally well-isolated, and are often used to representatively illustrate spectral changes. In Fig. 2, each alanine extract of a DARR spectrum is overlaid on the previous one in the cycle. The amino acid type and number as obtained from sequential assignments are indicated on the spectra, and identify the signals in terms of the Cα/Cβ atoms of the amino-acid they correlate in the two dimensions. Sequential assignment were obtained beforehand using 3D spectra which show much-reduced overlap when compared to 2D spectra, and extracts thereof are shown for the apo state in ref. [21,24] and for the ADP:AlF$_4^-$ and ADP:AlF$_4^-$:DNA complex in Supplementary Figures 4–6. Sequential assignments in NMR are obtained by using complementary sets of 3D spectra which correlate backbone atoms in an inter-residue manner, and allow to sequentially walk, i.e. connect, the backbone and Cβ atom resonances of the protein. Experiments for connecting side-chain resonances to the backbone complete the assignment process. Results are described in the Source Data file (for a detailed description of the assignment strategy see ref. [25,26] and for assignment details of all protein complexes studied herein Supplementary Table 1 and the Source Data file. The assignments for the apo state were extended compared to those described in ref. [21]). For the other compounds, the majority of the resonances could be assigned in a 3D spectrum by comparison with the two assigned compounds. In all cases, nucleotide binding is accompanied by chemical-shift changes (chemical-shift perturbations, CSP)[27], to a single new position (see Supplementary Figure 7), indicating efficient complex formation involving virtually all molecules in the sample, and the quasi-equivalence of all 6 NBDs.

Graphs of the site-specific $^{13}$C Cα-Cβ CSPs occurring in the CTDs between the two states obtained from 3D NCACB and NCACX spectra are shown in the third column. The overall strongest CSPs were observed for the AMP-PCP to ADP:AlF$_4^-$ step, the smallest CSPs for the apo to AMP-PCP step. As expected, some of the CSPs appear within the nucleotide-binding motifs (indicated by green background stripes in Fig. 2, third column). Interestingly, however, many CSPs occur outside of these motifs and particularly near the DNA-binding loops (residues 316Q-330Q and 355L-383D as putatively identified from the structural model of the apo protein[7] as the loops pointing to the interior of the protein, magenta stripes). Some resonances, including those in the loops of the NBD motifs, (see Supplementary Tables 2 and 3) appear or disappear in the different steps, and are indicated by pink and steel blue markers respectively in Fig. 2 on top of the CSPs (3rd column). When residues become observable in DARR spectra this either indicates large chemical-shift changes which prevent tracing the new to the original resonance, or residues reducing their dynamics and thus newly appearing in the NMR spectra. Indeed, MAS-NMR spectra are particularly sensitive to dynamics with correlation times around 1 µs that interfere with sample spinning and proton decoupling and lead to signal loss even for a small motional amplitude. While CSPs are indicative of conformational changes, appearing/disappearing resonances thus associate with dynamic

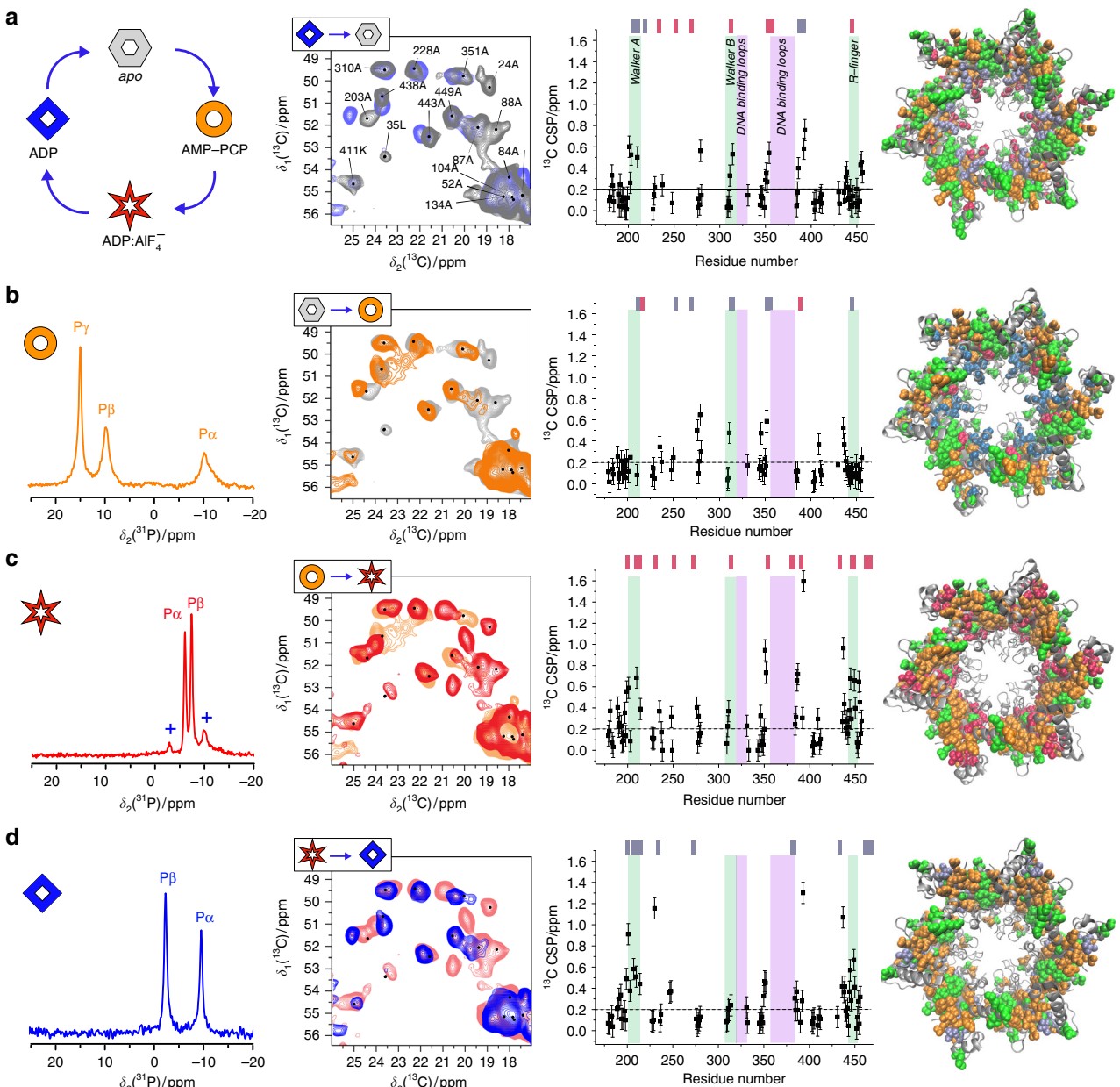

**Fig. 2** ATP hydrolysis equivalence scheme in the motor domain in the absence of DNA. Conformational changes in DnaB during the ATP hydrolysis cycle as shown in **a**, leftmost panel (see also Fig. 1b). Left column: $^{31}$P spectra of the three ATP analogues of **a**. +Denotes a minor contribution of the ADP complex shown in **d**. Second column: spectral fingerprints from the alanine Cα/Cβ region of $^{13}$C-$^{13}$C 20 ms DARR correlation spectra, compared to the ones of the previous step in the cycle (grey: apo, orange: AMP-PCP, red: ADP:AlF$_4^-$, blue: ADP). The spectrum of apo DnaB is taken from reference[24]. Third column: $^{13}$C Cα-Cβ CSPs (calculated according to $\mathrm{CSP} = \sqrt{\frac{1}{2}\left(\Delta\delta(^{13}C^\alpha)^2 + \Delta\delta(^{13}C^\beta)^2\right)}$) comparing again with the spectrum from the previous step. The dashed line represents a threshold above which we classify the CSPs as indicative for structural changes. The error bars (identical for all residues) are estimated to 0.1 ppm from the differences in the experimental spectra. The last column shows the $^{13}$C CSPs plotted on a model based on the X-ray structure of *Hp*DnaB complexed with ADP:Mg$^{2+}$ (residues 174–488 are shown, pdb 4ZC0, no electron density was observed for ADP:Mg$^{2+}$ [7]) (green: CSP < 0.2 ppm, orange: CSP ≥ 0.2 ppm, grey: unassigned). Newly appearing or disappearing signals are marked by pink or steel blue spheres, respectively, on the structure and their residue numbers are indicated by pink or steel blue bars on top of the CSP plots in the third column (for detailed residue numbers see Supplementary Tables 2 and 3). Known structural motifs are highlighted in the third column in green (Walker motifs and R-finger) and purple (DNA-binding motifs). All source data for the chemical-shift values of columns 1 and 3 are listed in the Source Data file

changes, and we equate appearing resonances with a stiffening of the corresponding residues. The large number of de novo peaks observed here, many well resolved in the 3D spectra[21,24] cannot be exclusively explained by large shifts, as this would involve an equal number of disappearing residues, which is not the case. It should be noted that these investigations were made in 3D spectra

(see Supplementary Figures 4−6) which show much less overlap than the 2D spectra of Fig. 2. It is highly unlikely that all disappearing peaks move to hide under peaks from other residues without leaving a clue. In particular, many of them belong to residues in loop regions (e.g., the Walker motifs or the DNA-binding loops) and can thus be found in characteristic and not

very crowded spectral regions. Stiffening of the corresponding amino acids is thus the most likely explanation for the large number of newly appearing resonances in the ADP:AlF$_4^-$ state and this state is clearly the one showing the highest rigidity.

The residues involved in the structural and dynamic rearrangements during the hydrolysis cycle are indicated on the *Hp*DnaB X-ray structure (only the CTD is shown) in column four of Fig. 2: residues with significant CSPs (≥0.2 ppm) are plotted in orange, those with minor CSPs (<0.2 ppm) in green, residues newly appearing in pink and disappearing ones in steel blue. One can see that not only residues near the nucleotide-binding site located at the interface between two DnaB monomers, but also ones further away are affected by the changes. This is most extreme for the NTD, not shown in the structural representation (Fig. 2, 4th column), of which all resonances that are seen in the apo spectrum surprisingly disappear when AMP-PCP and ADP are bound, but not for ADP:AlF$_4^-$ (Fig. 2, 2nd column). We attribute this highly interesting effect, currently subject to further studies, to changes in the dynamics of the NTD upon AMP-PCP and ADP binding.

**Single-stranded DNA binding**. To characterize the changes upon DNA binding, we added DNA to the different DnaB: nucleotide complexes representing the ATP-hydrolysis equivalence reaction. The binding of DNA to DnaB (three green arrows in Fig. 1b), results in the formation of ternary DNA-bound complexes referred to as AMP-PCP:DNA, ADP:AlF$_4^-$:DNA and ADP:DNA. We use a ssDNA substrate consisting of 20 thymine nucleotides, (dT)$_{20}$, which efficiently binds to nucleotide-bound DnaB[7,21].

Fluorescence anisotropy profiles of the different binding reactions of DnaB to the DNA are shown in Supplementary Figure 8a and reveal that in presence of ADP:AlF$_4^-$, ssDNA displays the smallest dissociation constant $K_d$ (Table 1, data in Supplementary Table 4), whereas in presence of ADP, DNA shows the poorest binding to DnaB. With either AMP-PCP or ATP, $K_d$ values were in-between these extremes. The apo form was not investigated as it does not bind ssDNA[21].

The left column of Fig. 3 shows $^{31}$P spectra of the three complexes. In all cases, two $^{31}$P resonance-lines in a 1:1 intensity ratio with different $^{31}$P chemical-shift values assigned to (dT)$_{20}$ are observed (see Supplementary Table 5), indicating that two thymine nucleotides bind per DnaB monomer in two slightly different $^{31}$P environments. The $^{31}$P resonances of the nucleotide remain visible after ssDNA binding, showing that the nucleotide remains bound. In all three cases, only one set of resonances for the bound nucleotide is observed, confirming the equivalence of all monomers in the oligomeric assembly. A second weak pair of resonances, Pα' and Pβ', belonging to DnaB:ADP without DNA (Fig. 3c; for the assignment see Supplementary Figure 8b–d) is seen in the ADP:DNA sample. We thus observe substoichiometric DNA binding in the presence of ADP as expected from the lower binding affinity (Table 1 and Supplementary Figure 8a). The $^{31}$P nucleotide Pα and Pβ chemical shifts of AMP-PCP and ADP change substantially upon ssDNA binding (see dashed lines in Fig. 3, Supplementary Figure 9 for $^{31}$P CSPs and for

AMP-PCP $^{31}$P-$^{31}$P 2D spectra), while the $^{31}$P CSPs are very small for ADP:AlF$_4^-$.

Analysis of the chemical shifts from $^{13}$C-$^{13}$C 20 ms DARR spectra in Fig. 3 (second column) indicates that the spectra of AMP-PCP, ADP:AlF$_4^-$, and ADP all change upon ssDNA binding (for larger spectral regions see Supplementary Figure 10), but to varying extents. Of interest here are the changes occurring in the CTD, for which the $^{13}$C Cα-Cβ CSPs upon ssDNA binding are plotted in the third column of Fig. 3. We also note the disappearance (ADP:AlF$_4^-$) or reappearance (AMP-PCP and ADP) of NTD signals upon DNA binding, indicating that the dynamics of the NTD is affected by both nucleotide and ssDNA binding within the CTD.

CSPs and appearing/disappearing residues are shown in the third column of Fig. 3. In the presence of AMP-PCP and ADP, many resonances show significant CSPs ≥ 0.2 ppm, which concentrate near the NBD motifs (Walker A and B and R-finger), and close to the DNA-binding loops. In addition, residues mainly in these regions newly appear in the spectrum upon binding (red bars on top of the CSP plots in Fig. 3). In the presence of ADP:AlF$_4^-$, the chemical shifts remain rather constant, and the few appearing residues are mainly located in the DNA-binding loops.

The rightmost column in Fig. 3 displays the changes on the structure of the DnaB-CTD. For AMP-PCP (Fig. 3a) and ADP (Fig. 3c), conformational changes are distributed over the protein structure indicating extensive rearrangements upon ssDNA binding. On the contrary, in presence of ADP:AlF$_4^-$, the protein conformation shows little changes around the nucleotide binding site upon DNA binding (Fig. 3b) suggesting that ADP: AlF$_4^-$ preorganizes the helicase into a state highly apt for ssDNA binding. This shows the tight coupling between the motor domain and the DNA-binding loops. Interestingly, in all nucleotide-bound states a major change observed upon DNA binding is the appearance of residues located in or near the DNA-binding loops (red residues in the rightmost column in Fig. 3). These residues line the central channel of DnaB through which the ssDNA passes. This indicates that this part of the protein stiffens upon DNA binding.

**DNA translocation**. The comparison of the three DNA-bound states allows to deduce information about the putative DNA translocation process (red reaction coordinate in Fig. 1). Figure 4 (first column) summarizes the $^{13}$C Cα-Cβ CSPs between adjacent states of the DNA translocation cycle for residues located in or close to the DNA-binding loop II (residues 355–383) (see Supplementary Figure 11a; for complete sequence-specific CSPs see Supplementary Figure 12 and Source Data file). The majority of residues in this loop are assigned in all three DNA-bound states and several show significant CSPs upon DNA binding.

Positively charged amino-acid residues are central in protein–DNA interactions[28]. The only lysine in the H4 motif of the DNA-binding loop (371D-382A)[29], 373K, as also determined by homology with DNA-bound *Bst*DnaB[10], experiences large $^{13}$C Cα-Cβ CSPs in the processes ADP:DNA → AMP-PCP:DNA and ADP:AlF$_4^-$:DNA → ADP:DNA (Fig. 4 and S9), which might be related to changes in sidechain–DNA interactions. Therefore, we recorded $^{15}$N-$^{13}$C 2D spectra of lysine sidechains (through setting the $^{13}$C frequency to the range of lysine side chain Cε carbon atoms, which resonate around 40 ppm) revealing peaks correlating Nζ and Cε atoms, and which should in case of interaction be sensitive to DNA binding, for the different compounds along the translocation pathway (Fig. 4, second column). For AMP-PCP: DNA and ADP:AlF$_4^-$:DNA the Nζ/Cε correlation peak of 373K is nearly at the same position, but it is shifting to an unresolved region or, more likely, is disappearing due to flexibility for the

**Table 1 ADP:AlF$_4^-$ is the best binder for DNA**

| | AMP-PCP: DNA | ADP:AlF$_4^-$: DNA | ADP:DNA | ATP:DNA |
|---|---|---|---|---|
| $K_d$/ nM | 58 ± 14 | 0.83 ± 0.03 | 5100 ± 500 | 14 ± 3 |

Overview of dissociation constants for the ssDNA bound complexes for the different nucleotide bound states along the motor cycle. The fit parameters are given in Supplementary Table 4

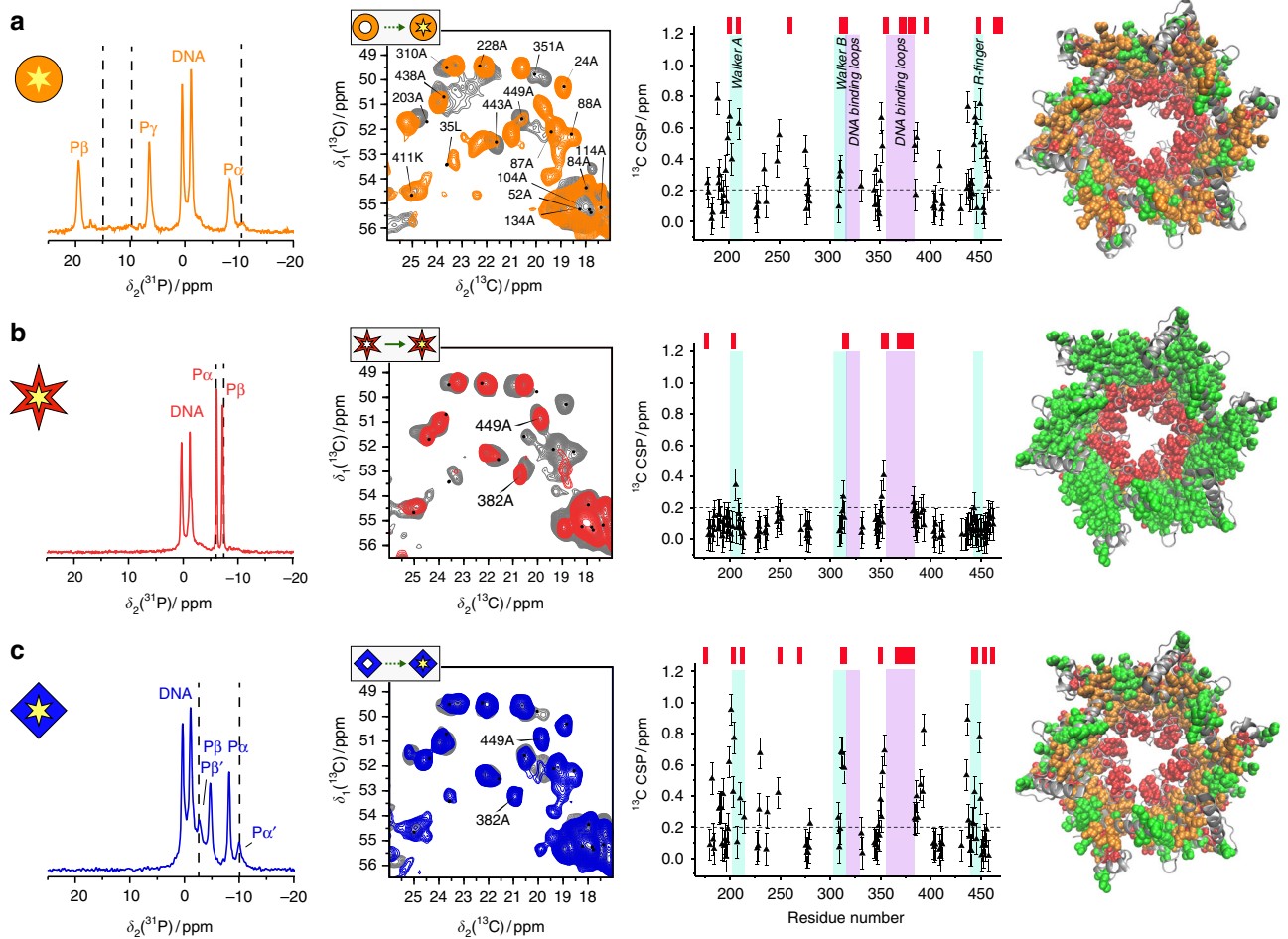

**Fig. 3** The DNA-binding process: ADP:AlF$_4^-$ reorganizes DnaB for DNA binding. $^{31}$P spectra (left column) of the ssDNA bound states of **a** AMP-PCP, **b** ADP:AlF$_4^-$, and **c** ADP. The dashed lines represent the $^{31}$P chemical-shift values in the absence of DNA, see also Fig. 2. Second column: $^{13}$C-$^{13}$C DARR correlation spectra with (coloured) and without (grey) ssDNA. Third column: $^{13}$C Cα-Cβ CSPs upon ssDNA binding. The error bars (identical for all residues) are estimated to 0.1 ppm from the differences in the experimental spectra. Residues highlighted in red on top of the CSP plot highlight resonances appearing in the NMR spectra upon DNA binding. The last column shows a plot of the $^{13}$C CSPs on a structural model based on pdb code 4ZC0 (residues 174–488 are shown) (green: CSP < 0.2 ppm, orange: CSP ≥ 0.2 ppm, red: newly appearing in the spectrum upon DNA binding, grey: unassigned). All source data for the chemical-shift values of columns 1 and 3 are listed in the Source Data file

ADP:DNA state. We speculate that this might be related to the binding and release of the lysine sidechain to the DNA during translocation: 373K bound for AMP-PCP and ADP:AlF$_4^-$, and released for ADP, as concluded from the shifting/disappearing correlation in the latter case. This interpretation is supported by the observation of $^{31}$P-$^{15}$N correlations between DNA and the sidechain of 373K in the ADP:AlF$_4^-$:DNA state (Supplementary Figure 13). Close by lysines, notably 209K (located in the Walker A motif) and 444K (next to the R-finger), also experience side-chain chemical-shift changes upon progression through the ATP hydrolysis cycle (see Fig. 4, second column). Considering arginines, we have identified 357R at the N-terminal end of a DNA-binding loop as a residue with considerable main-chain CSPs, and for which the sidechain resonances only appear upon DNA binding, as shown in the $^{15}$N-$^{13}$C sidechain correlation spectra, this time recorded for arginine sidechain atoms (by setting the $^{13}$C frequency range to around 160 ppm of the arginine side chain Cζ carbon atom), see Supplementary Figure 11b–d. They reveal 357R Nη1/η2/Cζ and Nε/Cζ correlations with large differences around 10 ppm in $^{15}$N chemical-shift values detected for the Nη1 and Nη2 atoms in 357R, which clearly indicate the participation of one of them, Nη2, in DNA binding.

The third column of Fig. 4 highlights the residues showing Cα-Cβ CSPs during DNA translocation on the X-ray structure of the CTD of DnaB, as well as the residues with appearing/disappearing resonances. The largest CSPs occur during the steps AMP-PCP:DNA → ADP:AlF$_4^-$:DNA and ADP:DNA → AMP-PCP:DNA.

Finally, we were interested in the fate of the Mg$^{2+}$ cofactor during the DNA translocation cycle. Therefore, we substituted Mg$^{2+}$ by Mn$^{2+}$, which induces paramagnetic relaxation enhancements (PREs) in the NMR spectra. Residues close to the metal centre (~10–15 Å for $^{13}$C) experience such strong PREs that they are broadened beyond detection and disappear in the NMR spectra[30,31]. Because of similar physical-chemical properties of Mg$^{2+}$ and Mn$^{2+}$ ions, the biological function is retained for DnaB[30,32]. The rightmost column of Fig. 4 shows the corresponding $^{31}$P spectra for the DNA-bound complexes with Mg$^{2+}$ and Mn$^{2+}$. For AMP-PCP:DNA (a) and ADP:AlF$_4^-$:DNA (b), the resonances of the bound nucleotide disappear due to the close spatial proximity to the metal ion, while the resonances of bound DNA experience only slight broadening effects. Interestingly, however, for ADP:DNA, the $^{31}$P resonances of the bound ADP in the DnaB:DNA-complex are still visible (Fig. 4c, last column).

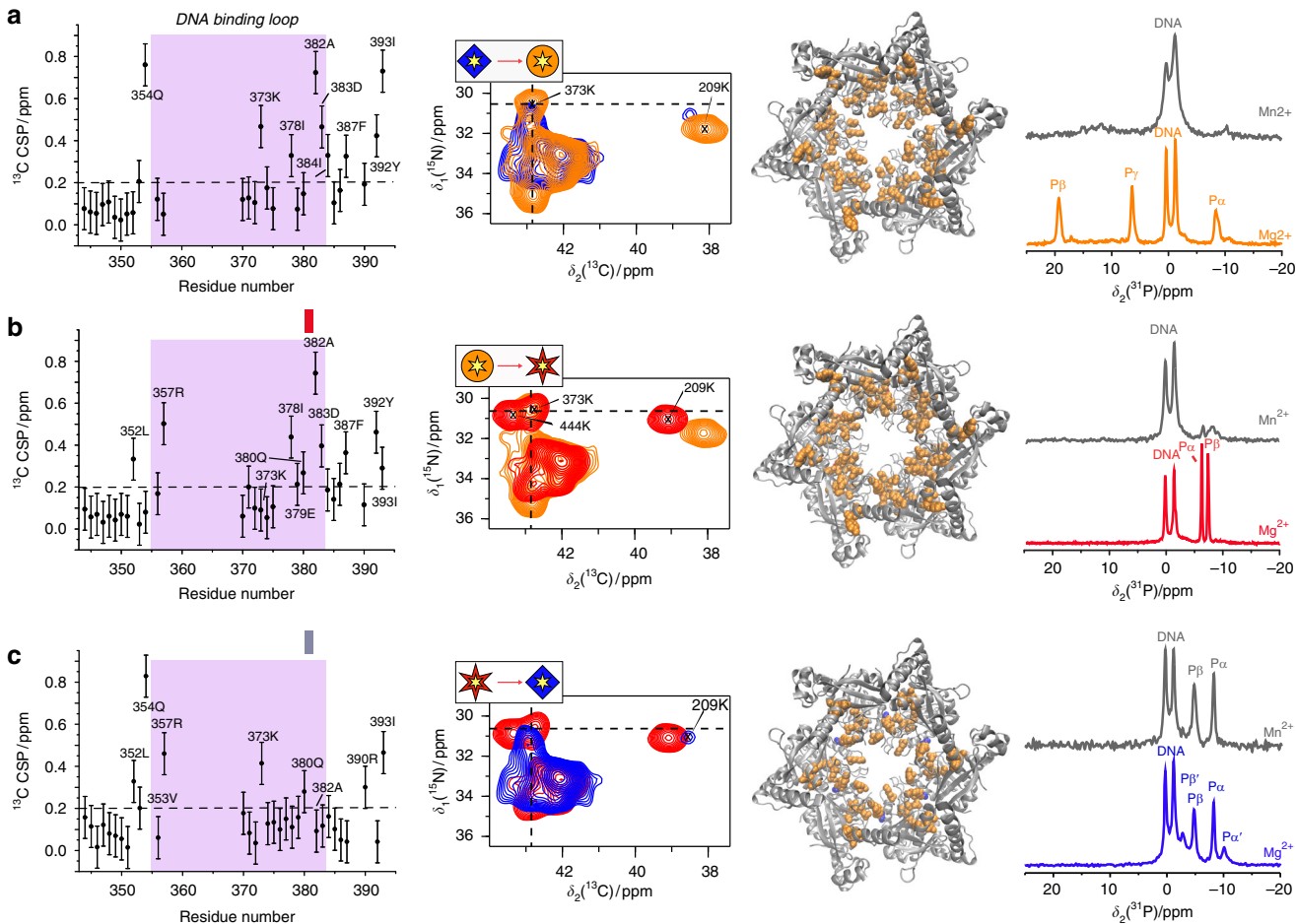

**Fig. 4** NMR allows to identify specific residues involved in DNA translocation. The first column shows the $^{13}C$ $C\alpha$-$C\beta$ CSPs for the indicated steps of the DNA translocation process (residues 344–393 are shown), the DNA-binding loop is highlighted in purple. The error bars (identical for all residues) are estimated to 0.1 ppm from the differences in the experimental spectra. $^{15}N$, $^{13}C$ correlation spectra (second column) for the lysine sidechains of the DNA-bound states for the three ATP-analogues as indicated. The third column shows a plot of the residues shown in the first column and experiencing CSPs > 0.2 ppm (orange) on a structural model based on pdb accession code 4ZC0. Red and steel blue colours indicate appearing and disappearing residues, respectively (see Supplementary Table 6 for a list). Residues with CSP < 0.2 ppm are not coloured. Right column: $^{31}P$ spectra of the DNA-bound complexes using $Mg^{2+}$ and $Mn^{2+}$ as metal ion cofactors indicating if the metal is bound (extinction of the ATP-analogue resonances for $Mn^{2+}$), or not (no change upon metal substitution). All source data for the chemical-shift values of columns 1 and 3 are listed in the Source Data file

Similar effects are seen in the carbon spectra (Supplementary Figure 14). This means that the metal ion is not present in this state. The strong $^{13}C$ CSPs in the Walker B motif during DNA translocation for the steps ADP:AlF$_4^-$:DNA → ADP:DNA and ADP:DNA → AMP-PCP:DNA (see Supplementary Figure 12) can thus be explained by the metal ion release and uptake respectively, which clearly changes the local electronic distribution for residues located in the Walker B motif coordinating the metal ion cofactor. Interestingly, in the absence of DNA, the $Mn^{2+}$ is not expelled in the corresponding step.

## Discussion
The use of ATP mimics has allowed us to characterize the sequence of arrested states appearing along the reaction coordinates indicated in Fig. 1a. Experimentally we have been able to produce and characterize seven of the eight mimics of Fig. 1b, only apo:DNA could not be prepared as the apo protein does not bind ssDNA[21]. From them, we could extract information about the structural and dynamic changes occurring during the nine reactions indicated by arrows in Fig. 1b. The assumption that the seven protein complexes forming the nodes of the network are a

good substitute for the elusive states appearing on the reaction coordinate is difficult to verify quantitatively as there is presently no method to characterize these elusive states experimentally; consequently, if we possibly do not investigate exactly the right states, we investigate the closest to right possible currently. This statement holds particularly for AMP-PCP for which it remains unclear to what extent the absence of a partial negative charge at the bridging methylene group influences its chemical properties. And indeed, the significant and coordinated changes that are observed in conformations (monitored by CSPs) and dynamics (leading to disappearing/appearing peaks) clearly couple the action at the nucleotide binding sites and DNA-binding loops. This indicates that this approximation clearly catches the major features of action of these motor proteins.

It is remarkable that DnaB binds ssDNA in the presence of ADP:AlF$_4^-$ more than an order of magnitude better than in the presence of ATP or its non-hydrolysable analogue AMP-PCP, and even several orders of magnitude better than ADP (Table 1). Concomitant with this, the most important change in the NMR spectra between all states of the ATP-hydrolysis cycle is induced when ADP:AlF$_4^-$ is bound to the protein. There are not only numerous conformational changes, but importantly many

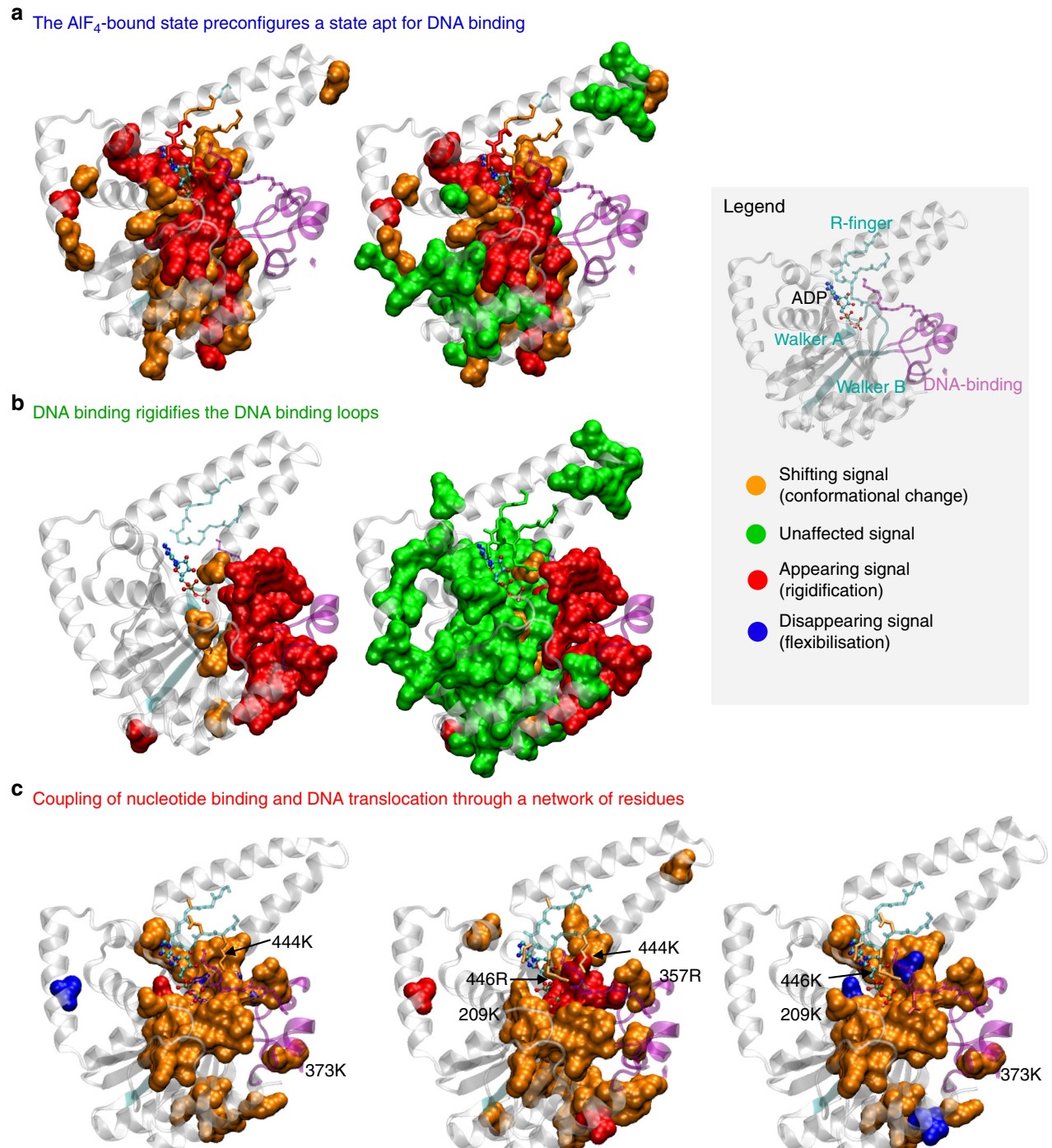

**Fig. 5** Site-specifically identified conformational change. **a** Details of conformational changes during ATP hydrolysis between the pre-hydrolytic state using AMP-PCP and the transition state mimicked by ADP:AlF$_4^-$, the left subpanel shows only the changing residues (see Legend box), the right subpanel in addition the unaffected residues; **b** changes upon DNA binding to the ADP:AlF$_4^-$bound state; and **c** changes during DNA translocation (ADP:DNA → AMP-PCP:DNA → ADP:AlF$_4^-$:DNA → ADP:DNA). The changes occurring at each arrow are shown in the three panels of **c**. The molecular structure shown represents a model in which the ADP is added based on homology with the *Aa*DnaB:ADP X-ray structure (pdb accession code 4NMN)[45]

residues stiffen in the presence of ADP:AlF$_4^-$ which were flexible in the previous step of the ATP hydrolysis reaction, in the presence of AMP-PCP. Figure 5 gives a detailed view of the CTD of a single DnaB monomer. The first row reports on changes between the AMP-PCP and ADP:AlF$_4^-$ bound states, and shows, in red, the residues that stiffen, in orange those which change conformation, and in green those remaining constant. In the post-hydrolytic state (Fig. 2d), many of these stiffened residues become

flexible again. These residues are mainly located around the NBD including the R-finger, as well as the Walker A and B motifs, and contact the DNA-binding loops via the Walker B motif and via the R-finger. The latter couples the ATP binding site to a DNA-binding loop of the adjacent monomer. The changes in the R-finger are emphasized in Supplementary Figure 15a, where one can see that 447N and 448G are stiffening upon binding of ADP:AlF$_4^-$, and that all remaining residues in the R-finger undergo

conformational changes in this step (in orange, Supplementary Figure 15a), including the highly conserved 446R. This residue is in close contact (3.2 Å in the model) with 382A that is part of the DNA-binding loop. The remainder of the assigned residues form a largely contiguous region not impacted by this hydrolysis step (green in Fig. 5b). Interestingly, the changes within a same monomer run through the central half-beta barrel, as displayed in Supplementary Figure 15b. This relatively rigid structural element might indeed be well-suited to perform the structural coupling between the motor unit and the DNA-binding loops. It is remarkable that dynamic effects play an essential role in this step, indicating a rather complex realization of the transmission element. The striking changes in the dynamics of the NTD upon nucleotide binding to the CTD give insight into the dynamic behaviour of this part of the protein located remotely from ATP and DNA-binding sites and are presently further investigated.

Fluorescence anisotropy measurements reveal that the presence of ADP:AlF$_4^-$ results in the strongest binding affinities of DNA to DnaB:nucleotide. While binding affinities are purely thermodynamic quantities, relating to the Gibbs free energy difference between bound and unbound state, the NMR data contribute information on the reaction mechanism: structural changes required for DNA binding are largely already prepared in the ADP:AlF$_4^-$-bound transition state analogue prior to DNA binding, so that the major change observed upon ssDNA binding (Fig. 5b, left) is the stiffening of the residues lining the central channel in the DnaB hexamer, and in particular the DNA-binding loops, while the bulk of the protein remains invariant (Fig. 5b, right). Arginine 357 (Fig. 3 and Supplementary Figure 11) at the beginning of the binding loop II interacts with DNA in all complexes (Fig. 4, left column) and is flexible in the structures without DNA. Likewise, the last part of the Walker B motif (315L-316Q) only appears upon DNA binding.

During the translocation steps, significant main chain CSPs are observed (Fig. 4). For this cycle, it is interesting to look in more detail at the lysine and arginine residues which are the preferential interacting residues with DNA[10,33,34]. Figure 5c paints the changes identified in Fig. 4 and Supplementary Figure 12 on a DnaB monomer: residues showing CSPs are drawn in orange, appearing residues in red, and disappearing residues in blue. 357R and 373K are both located in the binding loop II. Remarkably, even their backbone resonances show significant CSPs (Fig. 4). They both show conformational differences along the transport cycle (in its backbone as well as side chain resonances, Fig. 5c). 373K shows significant conformational changes in particular for the ADP:DNA → AMP-PCP:DNA and ADP:AlF$_4^-$:DNA → ADP: DNA steps. 357R instead experiences significant changes during the steps AMP-PCP:DNA → ADP:AlF$_4^-$:DNA and ADP:AlF$_4^-$: DNA → ADP:DNA (Fig. 5c). Note that 357R and 373K in HpDnaB correspond to 365R and 381R in BstDnaB which are in close spatial proximity to the DNA within the crystal structure of the BstDnaB–DNA complex[10]. These two residues, located at the beginning and end of loop II, are actually in close neighbourhood when considering a dimer. 444K is also changing its chemical shift considerably during the translocation cycle and is located close to the DNA-binding loop. 209K is at the end of the Walker A motif and sensitive to the changes in the ATP pocket occurring in the AMP-PCP:DNA → ADP:AlF$_4^-$:DNA step (Fig. 5c) as a part of a band of residues shown in red that rigidify in the transition-state conformation. We hypothesize that they are involved in transmitting the motor state to the DNA-binding loops.

Interestingly, our data indicate for all complexes studied a complete occupation of all six NBDs and a conformational equivalence of them. NMR chemical-shifts are highly sensitive to the local environments[35] and no resonance-splitting was observed neither in $^{13}$C- nor in $^{31}$P-detected spectra. Whenever a

resonance changes its position in the spectrum, it entirely shifts to its new position indicating that all monomers in the hexameric assembly behave similarly. This contrasts with the findings for BstDnaB:DNA for which a non-complete occupation of NBDs and a spiral staircase conformation was reported[10]. HpDnaB thus adopts under the experimental conditions used herein (e.g., an 18-fold excess of nucleotide compared to a DnaB monomer) a symmetric conformation with thus likely rather flat geometry in all complexes investigated. Whether the motor in vivo acts in a concerted or sequential manner during ATP hydrolysis cannot be judged from our data. However, they clearly indicate that symmetric DnaB structures can in principle be adopted by DnaB in presence of different nucleotide analogues. This is reminiscent of DNA-bound Rho[4] or E1[12] helicases in which the hexameric assembly adopts a flat conformation, but a spiral conformation is observed for the DNA-binding loops contacting the DNA. Our data are clearly consistent with such a model for HpDnaB.

Finally, we observe the release of the metal ion (Mg$^{2+}$) in the last step of DNA translocation (ADP:AlF$_4^-$:DNA → ADP:DNA), an event not described before to the best of our knowledge. This release does not happen in the cycle without DNA.

Our results highlight the important opportunity created by simple sedimentation[36–38] to produce complex preparations, allowing to sample a variety of points on reaction coordinates. This opens a new avenue for the investigation of reaction pathways not only of different classes of ATP-fuelled motor proteins, but also to follow other classes of proteins through their functional cycles.

## Methods

**Sample preparation.** AMP-PCP, ADP and ATP were purchased from Sigma-Aldrich and (dT)$_{20}$ from Microsynth.

**Expression and purification of HpDnaB.** The protein was cloned into the vector pACYC-duet1 (using the forward primer 5'-agtcatatggatcatttaaagcatttgcag-3' containing a NdeI restriction site and reverse primer 5'-atactcgagttcaagttgtaactatatca-taatcc-3' containing a XhoI site)[7], and expressed in the E. coli strain BL21 Star (DE3) (One Shot® BL21 Star™ (DE3) Chemically Competent E. coli, Invitrogen™). The overexpression was performed in M9 minimal medium[39] using $^{13}$C-enriched glucose 2 g L$^{-1}$ (Cambridge Isotope Laboratories, Inc. CLM-1396-PK) and $^{15}$N-enriched ammonium chloride 2 g L$^{-1}$ (Sigma-Aldrich® 299251) as sole carbon and nitrogen sources. The cells were lysed by a microfluidization process. $^{13}$C-$^{15}$N labelled HpDnaB was purified by heparin-agarose affinity chromatography using a 5 mL HiTrap Heparin HP column (GE Healthcare Life Sciences) followed by anion exchange chromatography using a 5 mL HiTrap Q HP column (GE Healthcare Life Sciences). The purified protein was concentrated up to 30 mg mL$^{-1}$ by centrifugation in buffer A (2.5 mM sodium phosphate, pH 7.5, 130 mM NaCl). For more details see ref. [37].

**Preparation of HpDnaB:ADP complexes.** 0.3 mM HpDnaB in buffer A was mixed with 5 mM MgCl$_2$ * 6H$_2$O and consecutively 5 mM ADP or ATP (~ 17-fold molar excess of nucleotide compared to an HpDnaB monomer) and incubated for 2 h at 4 °C. The protein solution was sedimented[36–38] in the MAS-NMR rotor (16 h at 4 °C at 210,000 × g) using home-build tools[40].

**Preparation of the HpDnaB:AMP-PCP complex.** 0.3 mM HpDnaB in buffer A was mixed with 5 mM MgCl$_2$ * 6H$_2$O and consecutively 15 mM AMP-PCP (~ 50-fold molar excess of AMP-PCP compared to an HpDnaB monomer) and incubated for 2 h at 4 °C. The protein solution was sedimented[36–38] in the MAS-NMR rotor (16 h at 4 °C at 210,000 × g) using home-build tools[40].

**Preparation of the HpDnaB:ADP:AlF$_4^-$ complex.** 0.3 mM HpDnaB in buffer A was mixed with 5 mM MgCl$_2$ * 6H$_2$O and consecutively 6 mM of an NH$_4$AlF$_4$ solution (prepared by incubating 1 M AlCl$_3$ solution with a 5-fold excess of 1 M NH$_4$F solution (compared to AlCl$_3$) for 5 min. in H$_2$O) and 5 mM ADP and incubated for 2 h at 4 °C. The protein solution was sedimented[36–38] in the MAS-NMR rotor (16 h at 4 °C at 210,000 × g) using home-build tools[40].

**Preparation of HpDnaB:nucleotide:DNA complexes.** The HpDnaB:nucleotide complexes were prepared as described above (in all cases 5 mM of nucleotide was used). 0.5 mM (for ATP) and 1 mM (ATP, AMP-PCP, ADP:AlF$_4^-$ and ADP) of

$(dT)_{20}$ was added to the complexes and reacted for 30 min at r.t. The protein solution was sedimented in the MAS-NMR rotor (16 h at 4 °C at 210,000 × *g*).

**Preparation of *Hp*DnaB:nucleotide: DNA complexes with $Mn^{2+}$.** The complexes containing $Mn^{2+}$ instead of $Mg^{2+}$ were prepared as described above, but instead of $MgCl_2$, $MnCl_2$ was used in a lower concentration (~1.5 mM corresponding to only a five-fold excess compared to a DnaB monomer). The lower concentration was used to avoid sizeable unspecific binding of $Mn^{2+}$ to the protein, e.g. to negatively charged residues[30].

**Fluorescence anisotropy measurements.** Equilibrium fluorescence anisotropy DNA-binding assays were performed on a Clariostar (BMG Labtech) microplate reader, fitted with polarization filters to measure fluorescence anisotropy. 0–8 μM DnaB (dodecamer) were titrated into 1 nM 5'-FAM labelled $(dT)_{20}$ (MWG Euro-fins) in binding buffer composed of 20 mM Hepes pH 7.5, 5 mM $MgCl_2$, 50 mM potassium acetate and 5% (v/v) glycerol and supplemented with 0.2 mg/ml BSA and 0.5 mM nucleotide (ATP, ADP, AMP-PCP or ADP:$AlF_4^-$ prepared in 50 mM Tris pH 8.0). After subtracting the polarization values obtained for DNA alone, the mean data from three independent experiments were fitted to a standard binding equation $Y = B_{max} * X^h/(K_d^h + X^h)$ assuming a single binding site with Hill slope ($h$) using GraphPad Prism6, where $Y$ is the difference between the anisotropy of completely bound and completely free oligo, $X$ is the DnaB concentration and $K_d$ is the equilibrium dissociation constant. The fits for ATP, AMP-PCP and ADP: $AlF_4^-$ were very good with $R^2$ values all above 0.99. For the ADP data, since the binding curves did not reach saturation, fitting of the data was achieved by constraining the $B_{max}$ value to 250, which is the value obtained for the other fits.

**Solid-state NMR experiments.** $^{13}C$ detected solid-state NMR spectra were acquired at 20.0 T static magnetic field strength using a 3.2 mm Bruker Biospin E-free probe[41]. The MAS frequency was set to 17.0 kHz. The 2D and 3D spectra were processed with the software TOPSPIN (version 3.5, Bruker Biospin) with a shifted (2.5–3.0) squared cosine apodization function and automated baseline correction in the indirect and direct dimensions. The sample temperature was set to 278 K[40]. All spectra were analysed with the software CcpNmr[42–44] and referenced to 4,4 −dimethyl-4-silapentane-1-sulfonic acid (DSS). The assignments of *Hp*DnaB are taken from reference 21 and extended in this work (see Source Data file). $^{31}P$-detected experiments were acquired at 11.74 T in a Bruker 3.2 mm probe using spinning frequencies of 12.0–17.0 kHz. The spectra were referenced to 85% $H_3PO_4$. All experimental details are provided in the Source Data file.

**Reporting summary.** Further information on experimental design is available in the Nature Research Reporting Summary linked to this article.

## Data availability

Data supporting the findings of this manuscript are available from the corresponding authors upon reasonable request. The following PDB structures were used in this study: 4ZC0 and 4NMN. The source data underlying Figs. 2, 3 and 4 and Supplementary Figs S7, S8a, S9a and S12 as well as the chemical-shift values for apo DnaB and DnaB:ADP:$AlF_4^-$:ssDNA and all experimental NMR parameters are provided as a Source Data file.

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

## Acknowledgements

This work was supported by the Swiss National Science Foundation (Grant 200020_159707 and 200020_178792), the French ANR (ANR-14-CE09-0024B), the LABEX ECOFECT (ANR-11-LABX-0048) within the Université de Lyon program Investissements d'Avenir (ANR-11-IDEX-0007), and the ETH Career SEED-69 16-1. This project has received funding from the European Research Council (ERC) under the European Union's Horizon 2020 research and innovation programme (grant agreement n° 741863, FASTER).

## Author contributions

T.W. collected the NMR data, J.T. performed and analysed the fluorescence anisotropy measurements, R.C. prepared the samples. T.W., D.L., A.B. and B.H.M analysed data, T.W., D.L., J.T., L.T, A.B. and B.H.M. interpreted. T.W., A.B. and B.H.M. designed research, A.B. and B.H.M. were the primary investigators. T.W., A.B. and B.H.M. wrote the paper which was edited and approved by all authors.

## Additional information

**Competing interests:** The authors declare no competing interests.

