## [Peer Review File · Nature Communications]

Reviewers' Comments:

Reviewer #1:

Remarks to the Author:

This is an excellent article describing the characterization of three states associated with the function of DnaB helicases, ATP hydrolysis, DNA binding, and DNA translocation. The authors generated nucleotide-, DNA-, and metal (Mg^{2+} and Mn^{2+}) bound states of the motor domain of the enzyme, and recorded P-31 and C-31 magic angle spinning solid-state NMR spectra, to infer the local structural and electronic environments in each of the states. They observe conformational and dynamic changes in the nucleotide binding domain that are connected to the binding and translocation of the DNA.

Overall, this study is of broad interest to the readership of Nature Communications. The work is carried out meticulously, and from the technical standpoint, the quality of the data is outstanding. Most importantly, there is a wealth of information that is available from this approach, which other structural techniques, such as X-ray crystallography or cryo-EM would not be able to provide. In this specific case, static structures would not inform on the changes in motions and electronic environments associated with the reaction coordinate, while solid-state NMR data provide a direct, atomic-level detail.

I recommend the publication of this work as is.

Reviewer #2:

Remarks to the Author:

This paper builds on previous work of the authors and uses NMR to study trapped intermediate states of DnaB helicases that use ATP hydrolysis to load and translocate DNA. It contains an impressive amount of NMR data obtained on 8 (!) different protein states that are arrested using nucleotide mimics and DNA fragments to sample the –in-principle- continuous conformational landscape of this process. To probe conformational changes the authors analyze chemical—shift changes in relationship to earlier structural and biochemical work. This work clearly merits publication in a high-ranking journal but the authors should address the following (minor) issues before acceptance:

[1] Assignments. Obviously, the current paper heavily relies on NMR assignments of the CTD domain. However, the information given in the ms on how these assignments were obtained is confusing and the actual values seem to be missing: The main text states on page 5: “Each spectrum is overlaid on the previous one in the cycle. Sequential assignments of the spectra were done, where possible, by transfer from the one of the apo state¹⁹, and were extended by further 3D experiments²⁰ as described in Table S1.

I could not find table S1, ref. 19 refers to 25% assigned residues and ref. 20 deals with a completely unrelated protein. In addition, what is the motivation to assume that the apo assignments can be transferred to the other states? Previous structural work predicts significant structural rearrangements that could strongly compromise such a strategy.

The SI stating that “The assignments of HpDnaB are taken from reference 11 and extended in this work (see caption Table S2)” isn’t really helpful either since the cited reference refers to the NTD and table S2 refers to 31P assignments. Hence, it is impossible to check and evaluate which and how many assignments have been obtained. I find this unacceptable and these data should be made available – at least for the reviewing process.

[2] Data analysis. Figure 3 reports chemical-shift changes for a subset of residues (130-450) between different states. Yet- the analysis is entirely based on the apo state. Indeed, chemical shift changes such as seen for the nucleotide binding region makes sense. Unfortunately, the rest of the discussion as well as the conclusions at the bottom of page 6 are difficult to understand.

For example, the authors write:

“While CSPs associate with conformational changes, appearing/disappearing resonances thus associate with dynamic changes, and we equate appearing resonances with a stiffening of the corresponding residues. The large number of de novo peaks observed here cannot be exclusively explained by large shifts, as this would involve an equal number of disappearing residues, which is not the case”

What about large structural rearrangements that lead to chemical-shift changes that have nothing to do with motion? Again, further information of the level of assignments is critical to draw reliable conclusions for the presented data. Did the authors conduct experiments at variable temperatures?

2nd example: "This is most extreme for the NTD, not shown in the structural representation (Figure 2, 4th column), of which all resonances surprisingly disappear from the apo spectra upon AMP-PCP and ADP binding, but not for ADP:AIF4- (Figure 2, 2nd column). We attribute this effect, currently subject to further studies, to changes in the dynamics of the NTD upon AMP-PCP and ADP binding."

Is this the case – why did the authors not present an analysis of the NTD? The changes in the CTD are indeed rather small and mostly related to ligand binding.

3rd example: "...interestingly, in all nucleotide-bound states a major change observed upon DNA binding is the appearance of residues located in or near the DNA binding loops ... This indicates that this part of the protein stiffens upon DNA binding and shows the tight coupling between the motor domain and the DNA binding loops".

Again – can you exclude structural as opposed to dynamic changes that can explain the appearance of new peaks? Why did the authors not attempt to actually probe the dynamics? – by for example measuring dipolar order parameters or by conducting relaxation studies ?

[3] Lastly, it is difficult to deduce from Figure 5 the novel aspects of this work compared to previous X-ray and EM work. Please clarify and provide a simpler figure. What do the authors mean by "conformational events"?

Also, using ATP mimetics is a widely applied strategy in X-ray, Cryo-EM and even NMR (See e.g. recent work by Glaubitz et al., Nat Comm) and should be appropriately cited.

Reviewer #3:

Remarks to the Author:

In the current manuscript, Wiegand et al. use solid state NMR to probe the structural response of the *H. pylori* DnaB helicase to different nucleotides and the binding of ssDNA. The authors find that the dynamics of different DnaB regions change depending on which type of ATP analogue is used and whether DNA is present. ADP-AIFx appears to particularly affect the pore loops of DnaB, which are responsible for translocation; biochemical analysis indicates that this analogue stabilises a state of the helicase that favors the tight binding of DNA.

Although the objective and approach of the study is of interest, the execution unfortunately falls short. The presentation of the work is overly technical and presumes a deep familiarity with solid state NMR and how to interpret data derived from such studies. There is little discussion of current views in the field as to how DnaB binds DNA and/or nucleotide, or how the current findings relate to such studies; there likewise are missed opportunities for expanding on certain aspects of DnaB dynamics or in testing some ideas generated by the data. Perhaps the most problematic issue is the assumption that the DnaB hexamer responds in a concerted, all-or-none manner as ATP is hydrolysed. It has been shown that the vast majority of ring-ATPases (going all the way back to the F1 ATPase) do not work by such a mechanism, and that different subunits in the hexamer instead adopt different ATPase state simultaneously. How the current data might be interpreted from this perspective is not discussed.

In summary, although the effort needed to collect the data presented here is well appreciated, the insights gained into DnaB mechanism are either largely confirmatory (e.g., that DnaB binds 2 nt/subunit, that pore residue dynamics change when DNA is present) or based on an overly simplistic view of the helicase's ATPase cycle. Publication in a more specialised journal would seem more appropriate.

Specific comments

1) The role of two residues (R357 and K373) in DNA translocation needs to be tested by both DNA binding and helicase assays.

2) The idea all subunits in a DnaB hexamer transit through the same ATPase states in synchrony with each other is likely incorrect. In crystallographic and EM-based studies of ring ATPases, there is clear pseudosymmetry in the system that is not reflected in the scheme shown for Fig. 1b or in the main text. This pseudosymmetry may well be quite difficult to detect by solid state NMR yet is probably a critical component of the ATPase cycle.

3) The authors seem to be missing out on an opportunity to expand and comment on the dynamics of the DnaB NTD, and how different regions of the NTD (the globular domain, the helical hairpin) respond to ATPase/DNA status. The mobility of this region and its response during DnaB cycling is only beginning to be understood

4) AMPPCP may or may not be a good ATP analogue. The absence of a partial negative charge on the bridging methylene group, as well as non-ideal bond angles at this position, can make AMPPCP serve as more of a product state (ADP-Pi) mimic in many cases, particularly on long time scales (minutes). Additional data are needed to show that AMPPCP really is ATP-like.

5) The data concerning the effects of ADP-AIFx would seem to accord well with structural data from the Steitz group; alternatively, is it possible that this analogue is not capturing a translocation state, but instead locking the helicase in a 'rigor' state that lies off-pathway to translocation? Please comment.

Minor points

P. 9. The assumption seems to be that all Lys373 residues will respond identically to a change in nucleotide state. This seems highly unlikely.

P. 9. Mn²⁺ is not a good substitute for Mg²⁺ for most ATPase reactions – it perturbs the hydrolysis reaction.

P. 12. Is the prediction that Arg357 and Lys373 bind to DNA borne out by the crystal structure from the Steitz group?

Fig. 2c. What do the "+" symbols means?

Fig. 5. Please label the different states on the figure itself.

Figure S2. This figure does not make sense as described. Was ATP only added to DnaB (green) or a mix of ATP and ADP? Also, what exactly is the time scale of the experiment? Finally, how can all the ATP be hydrolysed by the enzyme – are DnaB and ATP present at near equimolar concentrations (and allowed to incubate for hours)?

Fig. 2. It seems implausible that the authors can confidently assign specific amino acids to the resonance 'blob' shown in the lower right of the spectra panels.

The title for the legend for Fig. 2 is misleading – the data do not show that ATP hydrolysis occurs in the absence of DNA; no ATP is used and no release of product is followed.

Fig. S3. It is not obvious how the spectra shown support the claim of the legend title, namely that "Solid-state NMR allows to distinguish different states of ATP hydrolysis". No hydrolysis reaction is being followed in these measurements.

Reviewers' comments:

Reviewer #1 (Remarks to the Author):

I recommend the publication of this work as is.

Reviewer #2 (Remarks to the Author):

The authors should address the following (minor) issues before acceptance:

[1] Assignments. Obviously, the current paper heavily relies on NMR assignments of the CTD domain. However, the information given in the ms on how these assignments were obtained is confusing and the actual values seem to be missing: The main text states on page 5: "Each spectrum is overlaid on the previous one in the cycle. Sequential assignments of the spectra were done, where possible, by transfer from the one of the apo state¹⁹, and were extended by further 3D experiments²⁰ as described in Table S1.

I could not find table S1, ref. 19 refers to 25% assigned residues and ref. 20 deals with a completely unrelated protein. In addition, what is the motivation to assume that the apo assignments can be transferred to the other states? Previous structural work predicts significant structural rearrangements that could strongly compromise such a strategy.

Answer: Unfortunately, the reviewer did not obtain Tables S1 and S3 which (as requested for long Tables) were submitted separately. They are now called Supplementary Datasets 1 and 2 which are part of this submission and will also be made available for the readership. We have also inserted, following the reviewer's advice, a new Table (Supplementary Table 1) in the new numbering) with assignment statistics for all protein-complexes studied in this work. All the CSPs are listed in the Supplementary Dataset 2 also all the chemical-shift values allowing detailed tracking. Reference 23 (former reference 20) discusses the assignment strategy used. We have also clarified in the manuscript that we have significantly extended the assignment of (former) Ref 19 and have obtained the de novo sequential assignments on the apo protein as well as on the ADP:AlF₄⁻:DNA complex. The newly assigned apo resonances are now also listed in Supplementary Dataset 1.

The SI stating that "The assignments of HpDnaB are taken from reference 11 and extended in this work (see caption Table S2)" isn't really helpful either since the cited reference refers to the NTD and table S2 refers to 31P assignments. Hence, it is impossible to check and evaluate which and how many assignments have been obtained. I find this unacceptable and these data should be made available – at least for the reviewing process.

Answer: This is again caused by the fact that the reviewer did not obtain the info of Tables S1 and S3 (now Supplementary Datasets 1 and 2). Requested information is found in Supplementary datasets 1 and 2 as well as in Supplementary Table 1 for all constructs investigated. Note that the old Table S2 is now Supplementary Table 5.

[2] Data analysis. Figure 3 reports chemical-shift changes for a subset of residues (130-450) between different states. Yet- the analysis is entirely based on the apo state. Indeed, chemical shift changes such as seen for the nucleotide binding region makes sense. Unfortunately, the rest of the discussion as well as the conclusions at the bottom of page 6 are difficult to understand.

Answer: As detailed above, two forms, apo and DnaB:ADP:AlF₄⁻:DNA complex were sequentially assigned. This is now clearly stated in the manuscript and the reviewer has hopefully obtained the corresponding tables (now called Supplementary Datasets). The spectra of the other protein:ATP-mimic and protein:ATP-mimic:DNA complexes were assigned using the data of these two forms. We have also reformulated the text on the bottom of page 6.

For example, the authors write: “While CSPs associate with conformational changes, appearing/disappearing resonances thus associate with dynamic changes, and we equate appearing resonances with a stiffening of the corresponding residues. The large number of de novo peaks observed here cannot be exclusively explained by large shifts, as this would involve an equal number of disappearing residues, which is not the case”. What about large structural rearrangements that lead to chemical-shift changes that have nothing to do with motion? Again, further information of the level of assignments is critical to draw reliable conclusions for the presented data. Did the authors conduct experiments at variable temperatures?

Answer: We have discussed this point already in the original manuscript, but it was not clearly enough emphasized that the CSPs were obtained from 3D spectra not from 2D as shown in the Figures 2-4. The overlap is thus much lower as in the 2D spectra of the illustrations. Second, we argue here with the sheer number of disappearing resonances. It is highly unlikely that they all hide (in 3D!!) under peaks from other residues without leaving a clue. In particular, many of them belong to residues in loop regions (e.g. the Walker motifs or the DNA binding loops) and can thus be found in characteristic spectral regions. We have adapted the text in the middle of page 6 to make these points clearer. Temperature-dependent measurements were performed for the ADP-complex (at -20 °C and -193 °C), but unfortunately the disappearing peaks (when comparing to ADP:AlF₄⁻) could still not be observed. At the lower temperature, considerable line broadening is observed making interpretation difficult.

2nd example: “This is most extreme for the NTD, not shown in the structural representation (Figure 2, 4th column), of which all resonances surprisingly disappear from the apo spectra upon AMP-PCP and ADP binding, but not for ADP:AlF₄⁻ (Figure 2, 2nd column). We attribute this effect, currently subject to further studies, to changes in the dynamics of the NTD upon AMP-PCP and ADP binding.”

It this is the case – why did the authors not present an analysis of the NTD? The changes in the CTD are indeed rather small and mostly related to ligand binding.

Answer: Indeed, the mobility of this region is only beginning to be understood, and we have added in this work the important new element that the N-terminal domain can change its dynamic state as a function of nucleotide binding. We agree that this is indeed a highly interesting and new finding, and we are currently looking at this point in detail. Still, variable temperature experiments will not only make reappear the N-terminal domain, but also show the C-terminal domain, with the concomitant loss in resolution typically observed in NMR at sub-freezing temperatures. We have thus devised segmental isotope labelling of the NTD of DnaB in the full-length construct to uniquely observe the N-terminal domain (Wiegand et al., J. Biomol. NMR, 2018). Preliminary data exist, but this is a complex issue that warrants separate publication also because the function of the NTD is different from the functions discussed here (DNA binding and translocation).

3rd example: “...interestingly, in all nucleotide-bound states a major change observed upon DNA binding is the appearance of residues located in or near the DNA binding loops ... This indicates that this part of the protein stiffens upon DNA binding and shows the tight coupling between the motor domain and the DNA binding loops”.

Again – can you exclude structural as opposed to dynamic changes that can explain the appearance of new peaks? Why did the authors not attempt to actually probe the dynamics? – by for example measuring dipolar order parameters or by conducting relaxation studies ?

Answer: It would be interesting to perform such studies, but the dynamic residues (even when lowering the temperature drastically, see above) are just invisible in the spectra, precluding such measurements.

[3] Lastly, it is difficult to deduce from Figure 5 the novel aspects of this work compared to previous X-ray and EM work. Please clarify and provide a simpler figure. What do the authors mean by “conformational events”?

Answer: We changed “conformational events” by “conformational changes”. We have simplified the figure and have moved part of the info to Supplementary Figure 11. In response to referee 3 we also provide the labelling of the function directly on the figure.

Also, using ATP mimetics is a widely applied strategy in X-ray, Cryo-EM and even NMR (See e.g. recent work by Glaubitz et al., Nat Comm) and should be appropriately cited.

Answer: We agree and have added a reference in the manuscript, referring to the solid-state NMR work of Glaubitz and coworkers on ABC transporters (reference 15) and have added a further reference 16 regarding ATP-mimics (Bagshaw, C.R. ATP analogues at a glance. J. Cell Sci. 114, 459-460 (2001)).

Reviewer #3 (Remarks to the Author):

Specific comments

1) The role of two residues (R357 and K373) in DNA translocation needs to be tested by both DNA binding and helicase assays.

Answer: We have expressed both the single K373A and double R357A K373A mutants, and while proteins expressed well, they could not be purified using the protocols of the WT protein, as they likely aggregated on the column (see Figure of gels and purification on Q column).

Wild-type DnaB. The green vertical line represents ~50 % of buffer B (see SI).

Mutant DnaB R357A K373A. The green vertical line represents ~50 % of buffer B (see SI).

Mutant DnaB K373A.

In order to obtain additional direct evidence for the interaction between the DNA and K373, we recorded a NHPH spectrum on the DnaB:ADP:AlF₄⁻:ssDNA sample. The spectrum was recorded at 11.74 T at 17.0 kHz magic-angle spinning (measurement time 14 days). The spectrum clearly shows, by the presence of the peak labelled 373K, that the side-chain of Lys373 N ϵ contacts one of the two structurally distinct phosphate groups of the DNA, which in turn is in close proximity to amide nitrogens from neighboring residues D374, S375 and G376. This spectrum is a two-dimensional phosphorous-nitrogen correlation spectrum, in which the polarization evolves during a first period of time on the nitrogen spins, after cross polarization from the protons. The typical ¹⁵N frequencies of the spins can, after Fourier transform, be seen on the y axis. As K373 N ϵ has a unique ¹⁵N chemical shift, as determined by sequential assignments using 3D spectroscopy, it can be unambiguously identified. Even if the chemical shifts of the amide nitrogen spins of the neighbouring residues D374, S375 and G376 are not unique, the signals are fully consistent with an assignment to these amide nitrogens adding further evidence. After the transfer to the phosphorous spins by the use of adequate radiofrequency pulse sequences, the polarization is detected on the

phosphorous spins, and in the second dimension (x-axis), one can read the ³¹P frequency, of two different phosphate spins, of which only one is in close proximity (ca. 3-4 Å) to K373 N ϵ , since only the ³¹P around 0.5 ppm shows a cross signal with it. The other ³¹P at -1 ppm shows a correlation to D374, S375 and G376, and is thus not located far from the first one, but does not contact it directly. We have similar data for R357 for which we used fast magic-angle spinning (MAS) (> 100 kHz) to observe contacts between the arginine side chains and the DNA base. Fast MAS is a new approach which allows to observe proton resonances in solid-state NMR, remnant of solution NMR, through the averaging of the strong homonuclear dipolar interactions between protons. We are afraid that the inclusion of all these additional details would make the manuscript overly technical, as the referee already remarked. We thus refrain from including them in this contribution in technical details but just mention the presence of transfer and the experiments (page 9).

2) The idea all subunits in a DnaB hexamer transit through the same ATPase states in synchrony with each other is likely incorrect. In crystallographic and EM-based studies of ring ATPases, there is clear pseudosymmetry in the system that is not reflected in the scheme shown for Fig. 1b or in the main text. This pseudosymmetry may well be quite difficult to detect by solid state NMR yet is probably a critical component of the ATPase cycle.

Answer: NMR chemical shifts can normally detect pseudosymmetry in a much more sensitive manner than EM and x-ray at the resolutions obtained in the here-discussed objects. We have recently shown this at the example of the HBV capsid, which shows four distinct molecules in the unit cell, and where NMR detected the subtle structural differences in an atom-specific manner as chemical shift multiples in the spectra (Lecoq et al., *ChemBiochem* 2018). Our data on DnaB do not indicate any resonance splitting (neither in ¹³C or ³¹P spectra) pointing to (i) a complete occupation of all NBDs and (ii) a quasi-equivalence of all six NBDs. This indeed contrasts with the findings for

BstDnaB:DNA of Steitz et al., and might indicate that we look at a slightly different state. We have commented on that point in the Discussion section of the manuscript.

3) The authors seem to be missing out on an opportunity to expand and comment on the dynamics of the DnaB NTD, and how different regions of the NTD (the globular domain, the helical hairpin) respond to ATPase/DNA status. The mobility of this region and its response during DnaB cycling is only beginning to be understood.

Answer: Indeed, the mobility of this region is only beginning to be understood, and we have added in this work the important new element that the NTD can change its dynamic state as a function of nucleotide binding. We agree that this is indeed a highly interesting and new finding, and we are currently looking at this point in detail. Still, variable temperature experiments will not only make reappear the N-terminal domain, but also show the C-terminal domain, with the concomitant loss in resolution typically observed in NMR at sub-freezing temperatures. We have thus devised segmental isotope labelling of the NTD of DnaB in the full-length construct to uniquely observe the N-terminal domain (Wiegand et al., *J. Biomol. NMR*, 2018). Preliminary data exist, but thorough analysis thereof and its understanding in terms of the structural aspects are underway. The dissemination of what NMR can contribute to the central dynamics question of the N-terminal domain will be done as soon as possible. We have added a sentence to put this in perspective in our discussion (bottom of page 11)

4) AMPPCP may or may not be a good ATP analogue. The absence of a partial negative charge on the bridging methylene group, as well as non-ideal bond angles at this position, can make AMPPCP serve as more of a product state (ADP-Pi) mimic in many cases, particularly on long time scales (minutes). Additional data are needed to show that AMPPCP really is ATP-like.

Answer: It is certain that AMPPCP is the most ATP-like of the three different substitutes we studied, since both alternatives investigated, AMP-PNP and ATP γ S, got, on the timescale of NMR rotor filling (~16 hours) hydrolysed by DnaB in the presence of DNA (see Wiegand et al., *Angew. Chem. Int. Ed.*, 2016; a similar observation was made for a kinesin-type motor protein, see Suzuki, Y., Shimizu, T., Morii, H. & Tanokura, M. Hydrolysis of AMPPNP by the motor domain of ncd, a kinesin-related protein. *FEBS Lett.* 409, 29-32 (1997). It is an ongoing discussion if mimics are good mimics and which mimics are the best mimics, and all structural techniques are slave to this. As long as there are no highest-resolution X-ray structures for all different mimic states of a protein, or bond lengths and torsion angles are measured by other techniques, this question is hard to address. In this light, we agree that it is hard to find good mimics for the ATP-bound state, and that the best choice of analogue seems to depend very much on the system under study. We have included a discussion in the manuscript.

5) The data concerning the effects of ADP-AIFx would seem to accord well with structural data from the Steitz group; alternatively, is it possible that this analogue is not capturing a translocation state, but instead locking the helicase in a 'rigor' state that lies off-pathway to translocation? Please comment.

Answer: All mimics are by definition off-pathway to some degree, the only question remaining how far. There are no clear answers to this, as the pathway has not been characterized, and cannot with present methods, in detail. A strong indication that especially the ADP-AIF4 state is on-pathway is that it pre-organizes the helicase already in the absence of DNA binding in a conformation very close to the DNA-bound state. This might theoretically be pure coincidence, but very likely is not. We have also used the conditions Steitz et al. used (GDP:AIF4-) and obtained nearly the same state than for

ADP:AlF₄⁻. NMR can actually assess the typical geometry of ²⁷Al and ¹⁹F spins, and we have done NMR experiments supporting that AlF₄ adopts a (slightly distorted) squared-planar conformation as expected for the transition state of ATP-hydrolysis (to be published in a more technical Journal). Again, all NBDs are occupied with ADP:AlF₄ and show quasi-equivalence. It might well be that we are pushing the helicase artificially into certain conformations by using a large excess of ATP mimics. Again, this is valid for all structural studies, and we have mentioned that point in the discussion of our manuscript.

Minor points

P. 9. The assumption seems to be that all Lys373 residues will respond identically to a change in nucleotide state. This seems highly unlikely.

Answer: Our spectra indicate that resonances from all six K373 move during this process. No signal remains at the original position. Since we force every NBD to be occupied with the same ATP-mimic, this might be related to the sample preparation. Still, we note that our experimental conditions are close to those used in x-ray studies, i.e. (Itsathitphaisarn, O., Wing, Richard A., Eliason, William K., Wang, J. & Steitz, Thomas A. The Hexameric Helicase DnaB Adopts a Nonplanar Conformation during Translocation. *Cell* 151, 267-277 (2012), see also the SI) and in that work also all six DNA binding loops behave similarly (e.g. R381 coordinates the phosphate of the DNA, see below). We have added a discussion of this point at the end of our manuscript.

P. 9. Mn²⁺ is not a good substitute for Mg²⁺ for most ATPase reactions – it perturbs the hydrolysis reaction.

Answer: Mn²⁺ substitution can in some cases lead to reduced functionality, which is however not the case for *HpDnaB* (see reference 30: Soni, R.K., Mehra, P., Choudhury, N.R., Mukhopadhyay, G. & Dhar, S.K. Functional characterization of *Helicobacter pylori* DnaB helicase. *Nucleic Acids Res.* 31, 6828-6840 (2003).). Mg²⁺ substitution with manganese leads only to a reduction in DNA unwinding of ~20 % indicating that Mn²⁺ substitutes Mg²⁺ relatively well in our case. We have clarified this in the manuscript.

P. 12. Is the prediction that Arg357 and Lys373 bind to DNA borne out by the crystal structure from the Steitz group?

Answer: Initially these residues have indeed been identified as good candidates for DNA binding based on sequence alignment and homology modelling with existing structures (see also reference 27: Stelter, M. et al. Architecture of a Dodecameric Bacterial Replicative Helicase. *Structure* 20, 554 (2012)). In our work, we identified these residues directly in NMR experiments to receive polarization transfer from the DNA (see answer to “Specific comment 1” from Referee 3).

Fig. 2c. What do the "+" symbols means?

Answer: Sorry about the missing explanation in the caption- We added the information in the legend of Figure 2.

Fig. 5. Please label the different states on the figure itself.

Answer: Thanks for the suggestion, we have done so.

Figure S2. This figure does not make sense as described. Was ATP only added to DnaB (green) or a mix of ATP and ADP? Also, what exactly is the time scale of the experiment? Finally, how can all the ATP be hydrolysed by the enzyme – are DnaB and ATP present at near equimolar concentrations (and allowed to incubate for hours)?

Answer: We have clarified this point in the legend of Supplementary Figure 2. For the blue spectrum, only ADP was added for the green spectrum exclusively ATP. The incubation was performed for 2 h, the sedimentation takes 16 h. But even in the NMR rotor, the samples are fully hydrated and therefore ATP hydrolysis continues until all ATP is used up. This is what we observe in our spectra. 0.3 mM DnaB and 5 mM ADP or ATP respectively were used.

Fig. 2. It seems implausible that the authors can confidently assign specific amino acids to the resonance 'blob' shown in the lower right of the spectra panels.

Answer: All resonance assignments are based on a sequential walk based on three-dimensional NCACB, NCACX, CANCO, NCOCX as well as NcoCACB and CANcoCA experiments. The assignments are summarized in Supplementary datasets 1 and 2) which have unfortunately not been forwarded to the referees in the original submission. We give two references in the manuscript to the corresponding details. Also, chemical-shift perturbations were measured on 3D experiments where 2D was too overlapped. All experimental details are summarized in Supplementary Dataset 3. We have made this point clearer in the manuscript.

The title for the legend for Fig. 2 is misleading – the data do not show that ATP hydrolysis occurs in the absence of DNA; no ATP is used and no release of product is followed.

Answer: We have clarified that we study the equivalence scheme in Figure 2.

Fig. S3. It is not obvious how the spectra shown support the claim of the legend title, namely that "Solid-state NMR allows to distinguish different states of ATP hydrolysis". No hydrolysis reaction is being followed in these measurements.

Answer: We have clarified that we study the stationary points in equivalence scheme in Supplementary Figure 3.

Reviewers' Comments:

Reviewer #2:

Remarks to the Author:

The authors have clearly improved the manuscript, in particular by adding information regarding the use of 3D NMR. Since these data sets are absolutely critical for the conclusions drawn from the paper, I have only two requests before publication:

1. Ref. 22 only refers to the NTD. Representative 3D planes related to the main figures that helped assigning the CTD and improved the sequence coverage of the NTD should be presented in the SI.
2. It is not entirely clear how the authors defined the CSP of 0.2 ppm. What are the experimental ¹³C and ¹⁵N line widths? From visual inspection of the presented 2D data sets, ¹⁵N and ¹³C line width FWHH of at least 0.5 and 0.3 ppm, respectively, seem appropriate. The assignment tables give values with an 1/100 ppm accuracy. How reliable are these numbers?

Reviewer #3:

Remarks to the Author:

The revised work by Wiegand is improved, and addresses several issues raised on the prior round of review. Unfortunately, however, some particularly key points were not satisfactorily resolved. In particular:

1. There are still no supporting biochemical studies of the interactions and proposed model. Although alanine substitutions at R357 and K373 are said to be ill-behaved, the purification methods for these mutants are not discussed in the manuscript (and the only details regarding the native protein prep are ascribed to a previous publication). It is possible that the alanine mutants should be expressed/purified in a manner distinct from the WT protein – no discussion on this point is offered. More importantly, additional attempts with different substitutions (e.g., R357K/Q/M and K373Q/M) and/or altered purification strategies should be conducted, and the results from ATPase, DNA binding, and DNA unwinding experiments using any well-behaved mutants need to be conducted. The use of NHP spectra does not address the concerns central to this point; moreover, the decision to not include these highly technical and difficult to interpret results further highlights concerns about the general accessibility of the manuscript (point 3, below). Providing conclusive evidence for a functional role for key residues implicated from the NMR is necessary to support the claims of this study.
2. The flat-ring, all-or-none ATPase cycle proposed in paper still does not comport with states seen for nucleic acid-bound structures of hexameric helicases (e.g., Rho and E1). Relatively flat, symmetric rings are seen for DNA-free structures in some instances, such as large T-antigen and even DnaB, but these states are not considered to represent translocation intermediates. A new, 'concerted' translocation model would not necessarily be a problem, and would even be an exciting departure from what is expected, except that the scheme is not corroborated by accompanying biochemical studies or complementary structural methods.

It seems possible the data might accord with the spiral state imaged by the Steitz lab for GkDnaB – this protein has been imaged as a proposed translocation intermediate, in which all six active sites reside in a GDP-AlFx state. However, this structure may also represent a locked, off-pathway intermediate. This issue was raised previously, but the revision does not provide a clear answer.

Interestingly, recent studies of a number of other substrate-bound hexameric motors published in recent months (e.g., Hsp104, Yme1, etc.) have also revealed a strong asymmetry and non-planarity in their respective rings. The asymmetry of these systems turns out to be highly similar to Rho and E1 (and the F1 ATPase, which exemplifies asymmetric rotary motion), but is evident

only when both nucleotide and a translocation substrate are bound. Since the global conformational state of the HpDnaB ring here is inferred from local assignment data (as opposed to directly visualized), and since it is unclear whether spectra are being measured from DNA-bound protein (as opposed to DNA-free species that might represent a majority of the protein population), it is difficult to accept that the all-or-none model is on firm footing.

3. The manuscript is still extremely hard to follow for a non-NMR reader, which makes understanding the main findings of the study very challenging.

In sum, although the technical effort in the present work may have been heroic, the resultant insights into DnaB mechanism are not terribly compelling. As a consequence, recommendation in favor of publication in Nature Communications cannot be offered at this time.

Reviewer #2 (Remarks to the Author):

The authors have clearly improved the manuscript, in particular by adding information regarding the use of 3D NMR. Since these data sets are absolutely critical for the conclusions drawn from the paper, I have only two requests before publication:

1. Ref. 22 only refers to the NTD. Representative 3D planes related to the main figures that helped assigning the CTD and improved the sequence coverage of the NTD should be presented in the SI.

We have added Supplementary Figures 4 to 6 to illustrate the quality of these spectra.

2. It is not entirely clear how the authors defined the CSP of 0.2 ppm. What are the experimental ^{13}C and ^{15}N line widths? From visual inspection of the presented 2D data sets, ^{15}N and ^{13}C line width FWHH of at least 0.5 and 0.3 ppm, respectively, seem appropriate. The assignment tables give values with an 1/100 ppm accuracy. How reliable are these numbers?

The linewidth's in the ^{13}C dimensions (^{15}N shifts are not used for CSPs) are approximately FWHH = 0.4 ppm. A CSP in the order of the HWHH (0.2 ppm) can, at the given signal-to-noise ratio, easily be detected. The precision would even be slightly better but further systematic errors (e.g small inaccuracies in the sample temperature or differences in referencing) limit the precision. We agree with the referee that the accuracy of 1/100 ppm is too optimistic and have removed one digit. We now also state the estimated accuracy of 0.1 ppm.

Reviewer #3 (Remarks to the Author):

The revised work by Wiegand is improved, and addresses several issues raised on the prior round of review. Unfortunately, however, some particularly key points were not satisfactorily resolved. In particular:

1. There are still no supporting biochemical studies of the interactions and proposed model. Although alanine substitutions at R357 and K373 are said to be ill-behaved, the purification methods for these mutants are not discussed in the manuscript (and the only details regarding the native protein prep are ascribed to a previous publication). It is possible that the alanine mutants should be expressed/purified in a manner distinct from the WT protein – no discussion on this point is offered. More importantly, additional attempts with different substitutions (e.g., R357K/Q/M and K373Q/M) and/or altered purification strategies should be conducted, and the results from ATPase, DNA binding, and DNA unwinding experiments using any well-behaved mutants need to be conducted. The use of NHHP spectra does not address the concerns central to this point; moreover, the decision to not include these highly technical and difficult to interpret results further highlights concerns about the general accessibility of the manuscript (point 3, below). Providing conclusive evidence for a functional role for key residues implicated from the NMR is necessary to support the claims of this study.

Answer: while we have not performed new experiments (see also comment by the editor) we have decided to include the NHHP as Supplementary Figure 13 and provide, in the caption, an

explanation for the non-expert.

2. The flat-ring, all-or-none ATPase cycle proposed in paper still does not comport with states seen for nucleic acid-bound structures of hexameric helicases (e.g., Rho and E1). Relatively flat, symmetric rings are seen for DNA-free structures in some instances, such as large T-antigen and even DnaB, but these states are not considered to represent translocation intermediates. A new, 'concerted' translocation model would not necessarily be a problem, and would even be an exciting departure from what is expected, except that the scheme is not corroborated by accompanying biochemical studies or complementary structural methods.

We have extended the discussion on page 13 which now reads: “Interestingly, our data indicate for all complexes studied a complete occupation of all six NBDs and a conformational equivalence of them. NMR chemical-shifts are highly sensitive to the local environments³⁵ and no resonance-splitting was observed neither in ¹³C- nor in ³¹P-detected spectra. Whenever a resonance changes its position in the spectrum, it entirely shifts to its new position indicating that all monomers in the hexameric assembly behave similarly. This contrasts with the findings for *Bst*DnaB:DNA for which a non-complete occupation of NBDs and a spiral staircase conformation was reported¹⁰. *Hp*DnaB thus adopts under the experimental conditions used herein (e.g. an 18-fold excess of nucleotide compared to a DnaB monomer) a symmetric conformation with thus likely rather flat geometry in all complexes investigated. Whether the motor *in vivo* acts in a concerted or sequential manner during ATP hydrolysis cannot be judged from our data. However, they clearly indicate that symmetric DnaB structures can in principle be adopted by DnaB in presence of different nucleotide analogues. This is remnant of DNA-bound Rho⁴ or E1¹² helicases in which the hexameric assembly adopts a flat conformation, but a spiral conformation is observed for the DNA binding loops contacting the DNA. Our data are clearly consistent with such a model for *Hp*DnaB.”
The topic is introduced on page 3.

It seems possible the data might accord with the spiral state imaged by the Steitz lab for GkDnaB – this protein has been imaged as a proposed translocation intermediate, in which all six active sites reside in a GDP-AIFx state. However, this structure may also represent a locked, off-pathway intermediate. This issue was raised previously, but the revision does not provide a clear answer.

We were not sure whether the referee refers to GkDnaC (studied by Hsiao) which however is not using AIFx, or the Steitz *Bst*DnaB (also called GsDnaB) structure. In the latter, in contrast to our findings, only 5/6 NBDs are occupied with GDP-AIFx while we observe a complete occupation of all six NBDs. We have included a statement to make this point clearer (see text above).

Interestingly, recent studies of a number of other substrate-bound hexameric motors published in recent months (e.g., Hsp104, Yme1, etc.) have also revealed a strong asymmetry and non-planarity in their respective rings. The asymmetry of these systems turns out to be highly similar to Rho and E1 (and the F1 ATPase, which exemplifies asymmetric rotary motion), but is evident only when both nucleotide and a translocation substrate are bound. Since the global conformational state of the *Hp*DnaB ring here is inferred from local assignment data (as opposed to directly visualized), and since it is unclear whether spectra are being measured from DNA-bound protein (as opposed to DNA-free species that might represent a majority of the protein population), it is difficult to accept that the all-or-none model is on firm footing.

We can exclude that DNA-unbound protein exists as all isolated signals clearly shift/vanish completely. Signal/noise in these NMR spectra would be sensitive to detect about 5 % population, so it is evident that spectra are definitely being measured on the majority of the protein bound to the DNA. The spectra are measured from DNA-bound protein as can be seen from Figure 3 a+b from the ¹³C data (second row) where all the shifting and vanishing peaks (only grey) indeed shift or vanish completely upon DNA binding showing that no DNA-unbound protein exists. This is certainly true for the AMP-

PCP:DNA and ADP:AlF₄⁻:DNA complexes, whereas for ADP:DNA substoichiometric DNA binding in the presence of ADP as expected from the lower binding affinity is observed. Therefore, in the ADP:DNA case only ~ 2/3 of the protein are in the DNA bound state. This is clearly discussed in the manuscript (page 8).

3. The manuscript is still extremely hard to follow for a non-NMR reader, which makes understanding the main findings of the study very challenging.

We have made an effort to explain, for the non-specialist, the experiments used and the way they are interpreted.

In sum, although the technical effort in the present work may have been heroic, the resultant insights into DnaB mechanism are not terribly compelling. As a consequence, recommendation in favor of publication in Nature Communications cannot be offered at this time.